# Incidence, clinical course and risk factor for recurrent PCR positivity in discharged COVID-19 patients in Guangzhou, China: A prospective cohort study

**Jiazhen Zheng[1]☉, Rui Zhou[1]☉, Fengjuan Chen[2]☉, Guofang Tang[2], Keyi Wu[1], Furong Li[1], Huamin Liu[1], Jianyun Lu[3], Jiyuan Zhou[4], Ziying Yang[4], Yuxin Yuan[4], Chunliang Lei[2]\*, Xianbo Wu[1]\***

**1** Department of Epidemiology, School of Public Health (Guangdong Provincial Key Laboratory of Tropical Disease Research), Southern Medical University, Guangzhou, Guangdong, China, **2** Guangzhou Eighth People's Hospital, Guangzhou, Guangdong, China, **3** Department of Infectious Disease Control and Prevention, Guangzhou Center for Disease Control and Prevention, Guangzhou, Guangdong, China, **4** Department of Biostatistics, School of Public Health (Guangdong Provincial Key Laboratory of Tropical Disease Research), Southern Medical University, Guangzhou, Guangdong, China

☉ These authors contributed equally to this work.
\* gz8hlcl@126.com (CL); wuxb1010@smu.edu.cn (XW)

**Data Availability Statement:** Some data has been submitted at https://doi.org/10.5281/zenodo.

## Abstract

The phenomenon of COVID-19 patients tested positive for SARS-CoV-2 after discharge (redetectable as positive, RP) emerged globally. The data of incidence rate and risk factors for RP event and the clinical features of RP patients may provide recommendations for virus containment and cases management for COVID-19. We prospectively collected and analyzed the epidemiological, clinical and virological data from 285 adult patients with COVID-19 and acquired their definite clinical outcome (getting PCR positive or not during post-discharge surveillance). By March 10, 27 (9.5%) discharged patients had tested positive for SARS-CoV-2 in their nasopharyngeal swab after a median duration of 7·0 days (IQR 5·0–8·0). Compared to first admission, RP patients generally had milder clinical symptoms, lower viral load, shorter length of stay and improved pulmonary conditions at readmission (p<0.05). Elder RP patients ($\geq$ 60 years old) were more likely to be symptomatic compared to younger patients (7/8, 87.5% *vs.* 3/19, 18.8%, p = 0.001) at readmission. Age, sex, epidemiological history, clinical symptoms and underlying diseases were similar between RP and non-RP patients (p>0.05). A prolonged duration of viral shedding (>10 days) during the first hospitalization [adjusted odds ratio [aOR]: 5.82, 95% confidence interval [CI]: 2.50–13.57 for N gene; aOR: 9.64, 95% CI: 3.91–23.73 for ORF gene] and higher Ct value (ORF) in the third week of the first hospitalization (aOR: 0.69; 95% CI: 0.50–0.95) were associated with RP events. In conclusion, RP events occurred in nearly 10% of COVID-19 patients shortly after the negative tests, were not associated with worsening symptoms and unlikely reflect reinfection. Patients' lack of efficiency in virus clearance was a risk factor for RP result. It is noteworthy that elder RP patients ($\geq$ 60 years old) were more susceptible to clinical symptoms at readmission.

3988742. The rest is available within the manuscript and its Supporting Information files.

**Funding:** This work was supported by Open Project of Guangdong Provincial Key Laboratory of Tropical Disease Research. XW, JZ, RZ, FC, GT, FL, HL, KW, ZY, YY, JL, CL received the award. The grant numbers awarded to each author was 3,000 yuan. The full name of the funder is Fei Zou, https://baike.baidu.com/item/%E9%82%B9%E9%A3%9E/20392344?fr=aladdin. The funder had no role in study design, data collection and analysis, decision to publish, or preparation of the manuscript.

**Competing interests:** The authors have declared that no competing interests exist.

## Author summary

The baseline enrolled 285 patients admitted to Guangzhou Eighth People's Hospital (Guangzhou, Guangdong) with a diagnosis of COVID-19. We reported the epidemiology, clinical laboratory, radiological characteristics, virological results, treatment, and definite outcomes (getting PCR retested positive (RP) or not during post-discharge surveillance) of the cases. RP events occurred in nearly 10% of cases, were not associated with worsening symptoms and unlikely reflect reinfection. The lack of efficiency in virus clearance was a risk factor for RP result. Elder RP patients ($\geq 60$ years old) were more susceptible to clinical symptom at readmission. In the context of numerous COVID-19 cases showed SARS-CoV-2 positive again after discharged, the data in China may provide recommendations for post-discharge management, especially for other developing countries.

## Introduction

An outbreak caused by a novel human coronavirus, severe acute respiratory syndrome coronavirus 2 (SARS-CoV-2) was first detected in Wuhan in December 2019, [1] and has since spread within China and other countries. The WHO has declared the COVID-19 a pandemic on Mar 14, 2020.[2] As of May 19, 2020, more than four million confirmed cases and 315,131 deaths had been reported globally.[3]

So far, over tens of thousands of patients with COVID-19 have been clinically cured and discharged, but multiple COVID-19 cases showed SARS-CoV-2 positive again (redetectable as positive, RP),[4–9] which raises an attention for the discharged patients. Since RT-PCT testing for SARS-CoV-2 is known to have certain range of false negative rate [10], the false negative RT-PCR testing before discharging may be a reason for RP events. On the other hand, patients themselves may have certain characteristics that make them more vulnerable to being RP. Previously, Yao and colleagues conducted postmortem pathologic study in a ready-for-discharge COVID-19 patient (three consecutive PCR tests of nasopharynx swab samples showed negative results) who succumbed to sudden cardiovascular accident, found SARS-CoV-2 remained in lung cells which may account for the RP result in discharged COVID-19 patients.[11] However, more evidence is needed for addressing the following questions include what is the incidence of RP events? What are the clinical characteristics of RP patients before and after discharge? What are the risk factors for patients to get RP? The answers to these questions may lead to recommendations for clinical guideline for virus containment and discharge assessment. Therefore, to facilitate efforts on above questions, we prospectively collected and analysed detailed clinical data from adult patients with laboratory-confirmed COVID-19 and a definite clinical outcome (getting PCR positive or not during post-discharge surveillance) at Guangzhou Eighth People's Hospital, Guangzhou, China. In this study, we presented the clinical features of RP patients and explored the incidence and risk factors for RP events.

## Methods

### Study design and participants

This prospective cohort study included a cohort of adult inpatients ($\geq 18$ years old) from Guangzhou Eighth People's Hospital (Guangzhou, Guangdong) with a diagnosis of COVID-19. The diagnosis of COVID-19 was based on the New Coronavirus Pneumonia Prevention and Control Program (7th edition) published by the National Health Commission of China. [12] Overall, 285 patients who were admitted between January 20 and February 18, 2020 were

enrolled. March 10, all patients got a definite clinical outcome (becoming RP or haven't become RP during post-discharge surveillance). The final date of follow-up was March 14, 2020, the day all observed cases were discharged. This study was approved by the institutional ethics board of Guangzhou Eighth People's Hospital and the requirement for informed consent was waived by the ethics board.

## Data collection and processing

Both first and second hospitalization data including demographic information, epidemiological history, clinical signs and symptoms, underlying comorbidities, dynamic laboratory parameters, treatment measures and outcome data, were obtained from the electronic medical record system of Guangzhou Eighth People's Hospital by a trained team of experienced clinicians, epidemiologists and medical students using a standardized data collection form. According to the COVID-19 management routine of Guangzhou Eighth People's Hospital, after initial discharging, COVID-19 patients have to undergo a period of isolation (within 15 days after discharge, it is unlikely to have a RP after this time) in the hospital or at home. During the surveillance, nasopharyngeal swab samples of patients were collected by staff of Guangzhou Center for Disease Control and Prevention (Guangzhou CDC) and submitted to Guangzhou Eighth People's Hospital for Reverse transcription polymerase chain reaction (RT-PCR) test. Patients with positive nucleic acid tests after discharging were diagnosed as being RP and have to be readmitted and receive further medical observation. Two researchers (J.Z.Z. & R.Z.) independently reviewed and analysed the data and a third researcher (F.R.L.) adjudicated any difference in interpretation between the two primary reviewers.

## Testing process and analysis

Patients' nasopharyngeal swab specimens were collected for SARS-CoV-2 nucleic acid detection by RT-PCR at admission and once every two or three days during hospitalization and post-discharge surveillance. The detailed protocol of the RT-PCR is described elsewhere.[13] Threshold refers to the critical value of fluorescence signal in exponential growth period. Cycle threshold value (Ct value) refers to the number of cycles when the fluorescence signal reaches the threshold. A Ct-value less than 37 was defined as positive, a Ct-value ≥40 was defined as negative, and a medium load (Ct-value 37–40) was an indication for retesting.[14] Lower Ct value refers to higher viral load. Patients with positive nucleic acid tests in nasopharyngeal swab samples during post-discharge surveillance (within 15 days after discharge) were diagnosed as being RP. COVID-19 Human IgM IgG Assay Kit (ELISA based, produced by Abnova) was used to test the IgG and IgM level in COVID-19 patients.

## Discharge criteria for COVID-19

Individuals meeting the following criteria could be discharged: absence of fever for at least three days, substantial improvement in both lungs in chest CT, clinical remission of respiratory symptoms, and two throat-swab samples negative for SARS-CoV-2 RNA obtained at least 24 h apart.[15]

## Statistical analysis

Categorical variables are expressed as frequencies and percentages, and continuous variables are expressed as medians and interquartile ranges (IQRs). We compared the differences in epidemiological, clinical, and laboratory findings between patients who had a positive SARS-CoV-2 test after discharge and those who did not. Chi-square or Fisher's exact tests were used

to compare categorical variables between different patient groups, as appropriate, and the Mann-Whitney test was used to compare the continuous variables. When comparing the characteristics of RP patients between the two hospitalizations, the Wilcoxon signed-rank test and McNemar's test were applied, as appropriate. To evaluate the dynamic changes in laboratory tests, including Ct values, the median value of the first three weeks were compared between the RP and NRP patients. To explore the risk factors associated with being RP, univariate and multivariate-adjusted logistic regression models were used. In the multivariate adjusted model, age, sex, hypertension, diabetes and liver disease were adjusted.

All statistical analyses were performed using Stata SE, version 15 (StataCorp) and graphs were generated and plotted using GraphPad Prism version 8.00 software (GraphPad Software Inc). A $P$ value less than 0.05 (two-tailed) was considered statistically significant.

## Results

### Clinical data and laboratory findings during first hospitalization (RP *vs.* NRP patients)

From January 20 to March 4, 2020, 292 patients were admitted to Guangzhou Eighth People's Hospital. After excluding six patients who were minors ($\leq$18 years) and one death case, we enrolled 285 adult patients with COVID-19 in our final analysis. By March 14, 2020, all patients were discharged. Of these discharged patients, 27 (9.5%) recovered from COVID-19 tested positive for SARS-CoV-2 during post-discharge surveillance. The basic information is shown in Table 1. The median age of the study population was 48.0 years old (IQR 35.0–62.0, range, from 18.0–90.0 years), and 128 patients (44.9%) were men. The median length of stay (LOS) for both RP patients and NRP patients was 18 days. Generally, Demographics, epidemiological history and clinical symptoms did not significantly differ between the two groups.

The medians of RP and NRP patients' parameters during whole hospitalization and each week after admission are shown in Table 2. Compared with NRP patients, RP patients had a significantly lower median LDH level during hospitalization (159.5 *vs.* 186.0, p = 0.034). RP patients showed lower LDH (159.0 *vs.* 192.0, p = 0.034) than NRP patients at first week after admission, whereas eosinophil count was higher (0.05 *vs.* 0.02, p = 0.018). Concerning Ct values of N and ORF gene, there were no significant differences between the two groups within two weeks after admission. Eventually, RP patients' median Ct values of ORF gene were significantly lower than NRP group (35.5 *vs.* 39.0, p = 0.031) at third week. Similar results also observed in Ct values of N gene (Fig 1). The details of other markers between the two groups are described in S1 Table.

As for clinical course, RP and NRP patients' length of hospital stay (LOS) were both 18 days. For RP patients, the median duration of viral shedding (N gene) after admission was 14.0 days (IQR 8.0–20.0 days) (Fig 2), which was significantly longer than those in NRP patients (7.0 days [IQR 7.0–10.0]) (p<0.001). 62.9% RP patients and 23.6% NRP patients presented positive RNA detection tests (N gene) for more than 10 days since hospital admission. The results were similar in ORF gene (Table 3).

### Clinical data and laboratory findings of RP patients (first *vs.* second hospitalization)

After discharged, RP patients readmitted to hospital after a median of 7.0 days (IQR 5.0–8.0 days) of surveillance. Compared with the first hospitalization, more asymptomatic persons (17 [62.9%] *vs.* 5 [18.5%], p = 0.013), shorter length of hospitalization (7.0 days [5.0–11.0] *vs.* 18.0 [13.0–24.0], p<0.001) and higher Ct value of N gene (37.5 [36.0–38.5] *vs.* 35.0 [33.0–37.0],

**Table 1. Baseline characteristics of 27 RP patients and 258 non-RP patients.**

| Characteristics | All patients (n = 285) | RP Patients (n = 27) | NRP patients (n = 258) | p value |
|---|---|---|---|---|
| **Basic information** | | | | |
| Age, years | 48.0 (35.0–62.0) | 44.0 (32.0–62.0) | 49.0 (35.0–62.0) | 0.450 |
| Men | 128 (44.9) | 12 (44.4) | 116 (44.9) | 0.959 |
| Exposed to Wuhan or surrounding cities | 179 (62.8) | 19 (70.4) | 160 (62.0) | 0.393 |
| Smoking history | 31 (11.0) | 1 (3.7) | 30 (11.8) | 0.332 |
| Severity | | | | 0.703 |
| Mild | 22 (7.7) | 3 (11.1) | 19 (7.4) | |
| Moderate | 257 (90.2) | 24 (88.9) | 233 (90.3) | |
| Severe | 6 (2.1) | 0 (0) | 6 (2.3) | |
| Comorbidities | | | | |
| Any Comorbidity | 88 (31.4) | 8 (29.6) | 80 (31.0) | 0.832 |
| Hypertension | 51 (17.9) | 6 (22.2) | 45 (17.4) | 0.597 |
| Diabetes | 24 (8.4) | 1 (3.7) | 23 (8.9) | 0.712 |
| Liver disease | 23 (8.1) | 2 (7.4) | 21 (8.1) | 0.343 |
| COPD | 19 (6.7) | 2 (7.4) | 17 (6.6) | 0.698 |
| Cardiovascular disease | 18 (6.3) | 0 (0) | 18 (6.9) | 0.235 |
| Kidney disease | 8 (2.9) | 0 (0) | 8 (3.1) | .. |
| Cancer | 3 (1.1) | 0 (0) | 3 (1.2) | .. |
| **Clinical characteristic** | | | | |
| Symptoms | | | | |
| Asymptomatic | 34 (11.9) | 5 (18.5) | 29 (11.2) | 0.343 |
| Fever | 193 (67.7) | 18 (66.7) | 175 (67.8) | 0.902 |
| Dry cough | 159 (55.9) | 14 (51.6) | 145 (56.4) | 0.649 |
| Expectoration | 59 (20.7) | 6 (22.2) | 53 (20.5) | 0.838 |
| Chills | 58 (20.4) | 2 (7.4) | 56 (21.7) | 0.079 |
| Fatigue | 37 (12.9) | 4 (14.8) | 33 (12.8) | 0.764 |
| Myalgia | 34 (11.9) | 1 (3.7) | 33 (12.8) | 0.222 |
| Chest CT | | | | |
| Bilateral involvement of chest CT scan | 261 (95.6) | 27 (100.0) | 234 (95.1) | 0.241 |
| Small Pulmonary Nodules | 13 (4.7) | 3 (11.1) | 10 (4.1) | 0.125 |

Data are median (IQR) or n (%). p values comparing RP and NRP patients are from $\chi^2$, Fisher's exact test, or Mann-Whitney U test. COPD = Chronic obstructive pulmonary disease. CT = computerized tomography scan. RP = re-detectable as positive. NRP = non-re-detectable as positive.

p = 0.042) were presented in RP patients' rehospitalization (S2 Table). Elder RP patients ($\geq 60$ years old) were more likely to be symptomatic compared to younger RP patients (7/8, 87.5% vs. 3/19, 18.8%, p = 0.001) at readmission (Fig 3). Of those who underwent detection of the specific binding antibody to SARS-COV-2 in the plasma, twenty (100.0%) and sixteen (80.0%) showed positivity of IgG and IgM. During rehospitalization, duration of viral shedding from first positive tests (N gene) was 3.0 days (IQR 3.0–10.0 days) and 7.0 days (IQR 6.0–10.0 days) for ORF gene. 21 (77.8%) patients showed improved lung condition at readmission. (Table 3). The monitoring results of RNA for 27 RP patients during rehospitalization were shown in Fig 3 and S1 Fig.

Chest CT images during two hospitalizations of three RP patients and CT images taken at two different time during first hospitalization of one NRP patient were shown in Fig 4. The typical findings of chest CT images of RP patients at readmission were the improved bilateral pulmonary inflammation.

**Table 2. Laboratory indicators of RP and NRP patients for the first three weeks of hospitalization.**

| Parameter, week | Normal Range | All patients | RP Patients | NRP patients | p value |
|---|---|---|---|---|---|
| Positive Ct value (N) | ≥40 | 37.0 (34.5–38.0) | 35.5 (34.0–37.0) | 37.0 (35.0–38.0) | 0.044 |
| 1 | | 36.0 (33.0–38.0) | 35.0 (33.0–36.0) | 36.0 (33.0–38.0) | 0.197 |
| 2 | | 37.0 (34.0–38.8) | 35.8 (31.8–38.3) | 37.0 (34.5–39.0) | 0.275 |
| 3 | | 38.0 (33.0–39.0) | 36.0 (33.0–38.0) | 38.0 (37.0–39.0) | 0.045 |
| Positive Ct value (ORF1ab) | ≥40 | 38.0 (35.0–39.0) | 37.0 (34.0–38.0) | 38.0 (35.5–39.0) | 0.061 |
| 1 | | 37.0 (34.0–38.0) | 35.0 (33.0–37.5) | 37.0 (34.0–39.0) | 0.088 |
| 2 | | 38.0 (34.0–39.0) | 36.3 (33.5–39.0) | 38.0 (35.0–39.0) | 0.224 |
| 3 | | 38.0 (36.0–40.0) | 35.5 (33.0–39.0) | 39.0 (38.0–41.0) | 0.031 |
| White blood cell count, ×10$^9$/L | 3.5–9.5 | 5.3 (4.4–6.3) | 5.2 (4.2–5.6) | 5.3 (4.4–6.4) | 0.301 |
| 1 | | 4.9 (3.9–6.3) | 4.8 (4.2–5.5) | 5.1 (3.9–6.3) | 0.297 |
| 2 | | 5.4 (4.5–6.8) | 5.1 (4.6–5.7) | 5.6 (4.5–6.8) | 0.246 |
| 3 | | 5.5 (4.6–6.5) | 5.4 (4.4–6.3) | 5.5 (4.6–6.5) | 0.489 |
| Eosinophil, ×10$^9$/L | 0.02–0.52 | 0.08 (0.04–0.12) | 0.08 (0.04–0.12) | 0.08 (0.04–0.12) | 0.767 |
| 1 | | 0.03 (0.0–0.07) | 0.05 (0.02–0.1) | 0.02 (0.00–0.07) | 0.018 |
| 2 | | 0.09 (0.05–0.14) | 0.08 (0.04–0.10) | 0.1 (0.05–0.1) | 0.208 |
| 3 | | 0.12 (0.08–0.20) | 0.11 (0.08–0.17) | 0.12 (0.08–0.20) | 0.729 |
| Lactate dehydrogenase, U/L | 125–243 | 185.0 (154.5–225.0) | 159.5 (139.0–197.0) | 186.0 (155.5–229.0) | 0.034 |
| 1 | | 188.0 (151.0–238.0) | 159.0 (140.0–196.0) | 192.0 (152.0–243.0) | 0.022 |
| 2 | | 183.0 (149.0–238.0) | 159.5 (147.0–213.0) | 184.0 (151.0–238.0) | 0.493 |
| 3 | | 179.0 (150.0–213.5) | 166.0 (129.0–192.0) | 180.0 (151.0–218.0) | 0.069 |
| C-reactive protein, mg/L | <10 | 5.0 (5.0–12.7) | 5.0 (5.0–5.0) | 5.0 (5.0–13.4) | 0.094 |
| 1 | | 5.0 (5.0–23.5) | 5.0 (5.0–10.2) | 5.0 (5.0–26.2) | 0.034 |
| 2 | | 5.0 (5.0–11.5) | 5.0 (5.0–5.0) | 5.0 (5.0–12.3) | 0.301 |
| 3 | | 5.0 (5.0–5.0) | 5.0 (5.0–5.0) | 5.0 (5.0–5.0) | 0.537 |

Data are median (IQR) value of first three weeks after admission., the number of available test result of RP patients for the first, second and third weeks were 27, 27, 19, in contrast, 258, 249, 184 in NRP patients. P values comparing RP and NRP patients are from Mann-Whitney U test. RP = redetectable as positive. NRP = non-redetectable as positive.

## Risk factors for RP events

In the univariate logistic regression model, decreased median Ct values of ORF gene at week three after admission (OR 0.76, 95% CI 0.60–0.97) and duration of viral shedding from admission greater than 10 days (OR 5.49, 95% CI 2.39–12.62 and OR 8.77, 95% CI 3.64–21.09, for N gene and ORF gene, respectively) were associated with increased risk of being RP. When adjusting for age, sex, hypertension, CVD and liver disease, our regression model showed similar results (Table 4).

## Discussion

This study reported the incidence rate of and risk factors for RP events in adult patients with COVID-19 in Guangzhou. Additionally, the epidemiological, clinical and virological features of RP and NRP patients were compared. The viral load in both RP patients and NRP patients' nasopharyngeal swab samples were monitored with sustained viral detection by RT-PCR. As of March 14, 2020, the end of the follow-up, 27 (9.5%) patients had become RP during their post-discharge surveillance after a median duration of 7.0 days. We revealed that a longer duration of viral shedding and higher viral load in the later stage of hospitalization were risk

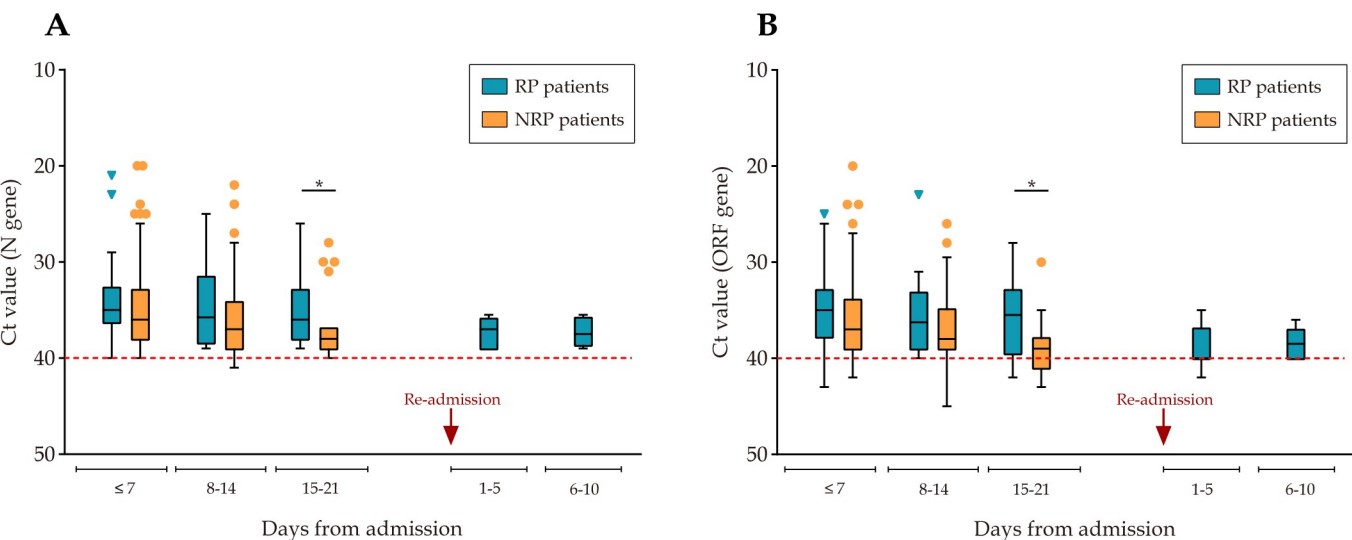

**Fig 1. Comparison of viral dynamics between RP and NRP patients.** Figure shows temporal changes in median Ct value of N gene (A) and ORF gene (B) in different time period. Since we have only collected and analyzed the data during patients' hospitalization, Ct value at admission and readmission were unavailable. The dotted line in red shows the lower normal limit of Ct values. Ct = cycle threshold. RP = redetectable as positive. NRP = non-redetectable as positive.

factors for RP events in patients with COVID-19. Furthermore, our study found that elder RP patients ($\geq$ 60 years old) were more susceptible to clinical symptom at readmission.

27 of 285 (9.5%) individuals had tested positive for SARS-CoV-2 in nasopharyngeal swab after discharged. Previously, Zhongnan Hospital has reported two (3·23%) medical staff had tested positive after discharged.[7] In a study on 209 discharged patients, conducted by Tang et al [16], 9 (4.3%) re-tested positive in throat swabs only, 13 patients (6.2%) re-tested positive

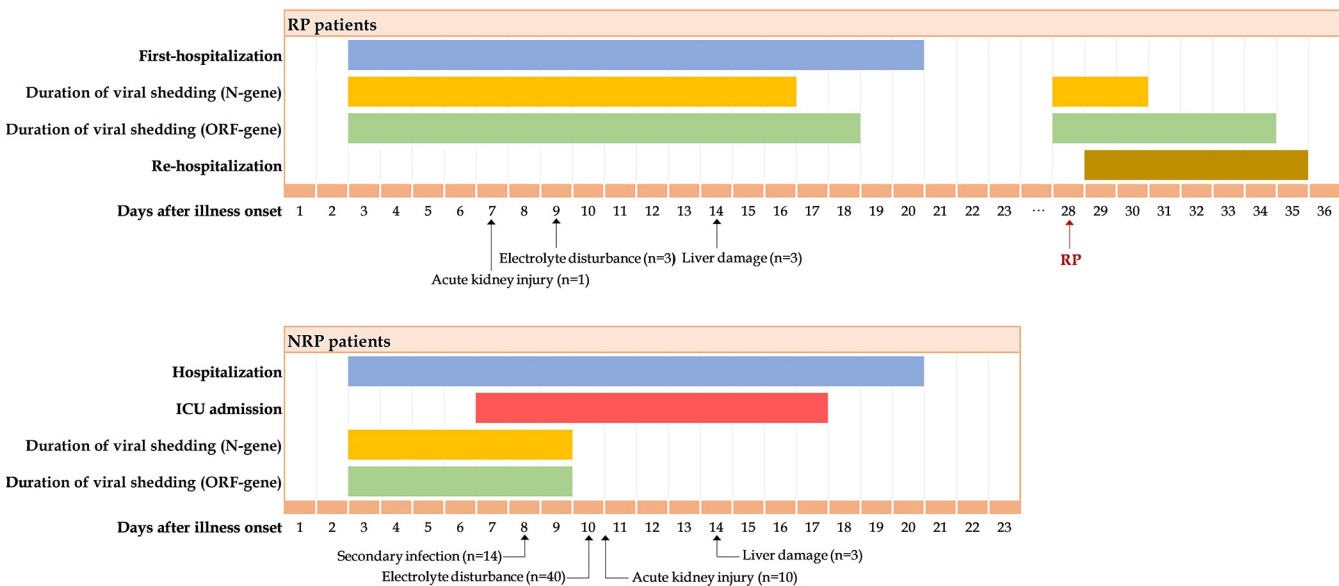

**Fig 2. Clinical course, complications and duration of viral shedding from illness onset in patients hospitalized with COVID-19.** Figure shows median duration of hospitalization and positive nucleic acid Ct value and onset of several complications. RP = redetectable as positive. NRP = non-redetectable as positive.

**Table 3. Clinical course and RNA test result of 27 RP patients and 258 non-RP patients.**

|  | RP (n = 27) | NRP (n = 258) | p value |
|---|---|---|---|
| **First-hospitalization** |  |  |  |
| Time from illness onset to admission, days | 3.0 (1.0–5.0) | 3.0 (1.0–7.0) | 0.923 |
| Length of stay, days | 18.0 (13.0–24.0) | 18.0 (13.0–25.0) | 0.822 |
| Duration of viral shedding (N) after admission, days | 14.0 (8.0–20.0) | 7.0 (7.0–10.0) | <0.001 |
| Distribution, no (%) |  |  |  |
| ≤10 days | 10 (37.0) | 197 (76.4) | <0.001* |
| >10 days | 17 (62.9) | 61 (23.6) |  |
| Duration of viral shedding (ORF) after admission, days | 16.0 (8.0–21.0) | 7.0 (7.0–10.0) | <0.001 |
| Distribution, no (%) |  |  |  |
| ≤10 days | 8 (29.6) | 203 (78.7) | <0.001* |
| >10 days | 19 (70.4) | 55 (21.3) |  |
| **Rehospitalization** |  |  |  |
| Quarantine site before rehospitalization |  |  |  |
| Hospital | 9 (33.3) | - | - |
| Home | 18 (66.7) | - | - |
| Time from discharge to retest positive, days | 7.0 (5.0–8.0) | - | - |
| Length of stay, days | 7.0 (5.0–11.0) | - | - |
| Duration of viral shedding after being RP (N gene), days | 3.0 (3.0–10.0) | - | - |
| Duration of viral shedding after being RP (ORF gene), days | 7.0 (6.0–10.0) | - | -. |
| Lung inflammation compared with first hospitalization |  |  |  |
| Normal | 1 (3.7) | - | - |
| Improved | 21 (77.8) | - | - |
| Stable | 5 (18.5) | - | - |
| Aggravated | 0 | - | - |

Data are median (IQR) or n (%). P values comparing RP and NRP patients are from $\chi^2$, Fisher's exact test, or Mann-Whitney U test. ICU = intensive care unit.

RP = redetectable as positive. NRP = non-redetectable as positive.

*$\chi^2$ test comparing all subcategories.

in anal swabs only, and 22 (10.5%) re-tested positive in either. The study by Zhongnan Hospital merely included medical staff and the Third People's Hospital of Shenzhen had only enrolled 49.4% (209/423) COVID-19 patients diagnosed in Shenzhen. The incidence rate of our study was more representative since we have included the majority (75.6%, 285/377) of COVID-19 patients in the international city: Guangzhou, with no restrictions other than age. Since the outbreak of COVID-19 occurred earlier in China than in other countries and the outbreak is still increasing or plateauing in many other countries,[3] the incidence rate reported in this study may provide a reference for the global disease management, especially in the populous developing countries.

Retested positivity unlikely reflect reinfection, since most RP patients in current study showed no obvious clinical symptoms or disease progression indicated by laboratory and CT findings and did not contact with other infectious patients. When we focus on chest CT rather than viral load, most of the RP patients seem to be normal convalescent patients with favorable inflammatory absorption. For these patients, unless there is a clinical symptom worsening, excess clinical intervention may not be necessary. It has been shown that the plasma levels of the IgM and IgG antibodies specific to SARS-CoV-2 and the series of immune cells produced during recovery play important roles in virus neutralization and prevention against further infection [17,18]. In a study of rhesus macaques infected with SARS-CoV-2, the animals did

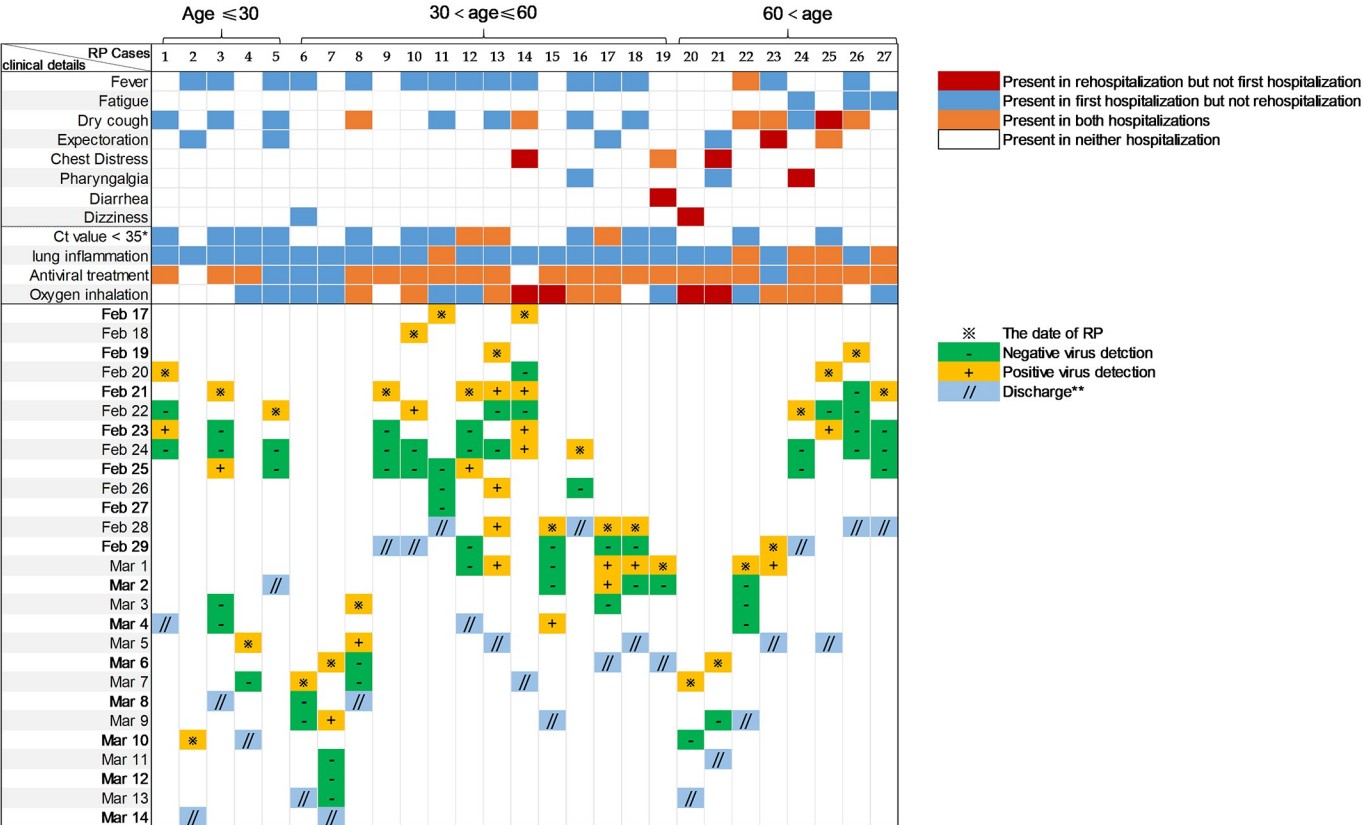

**Fig 3. Comparison of the two hospitalization courses of 27 RP patients and result of series SARS-CoV-2 RNA test in nasopharyngeal swab specimens during the second hospitalization.** Comparisons of clinical condition between first and second hospitalization are shown for each RP patient (upper panels). Timeline of series SARS-CoV-2 RNA test (lower panels) during rehospitalization are shown. *Ct value <35 refers to whether the lowest Ct value during hospitalization is lower than 35. **Discharge indicates two throat-swab samples negative for SARS-CoV-2 RNA obtained at least 24 h apart. This figure showed that elder RP patients ($\geq$ 60 years old) were more likely to be symptomatic compared to younger RP patients (7/8, 87.5% vs. 3/19, 18.8%, p = 0.001) at readmission. RP = redetectable as positive. NRP = non-redetectable as positive.

not develop reinfection following recovery and re-challenge [19]. Thus, we concluded that RP events are more likely caused by false negative RT-PCR tests before discharging.

Concerning clinical symptom were more commonly seen in elder RP patients. Previous studies found that age-dependent defects in T-cell and B-cell function and the excess production of type 2 cytokines could lead to more prolonged proinflammatory responses and constant clinical symptom.[20]. On this basis, we speculate that the remained clinical symptoms may be related to the poor recovery ability and prolonged body responses of the elderly just recovered from COVID-19. Thus, enhanced follow up medical examination and treatment should be carried out in time for discharged elderly patients.

In multiple respiratory viruses, viral load could be a predictor of disease stage and progression.[20–23] In this study, Ct values of respiratory tract samples from both RP and NRP patients with COVID-19 peaked in the first week after admission which was similar to the results reported in Beijing,[24] but distinct from those observed in patients with SARS, which normally peaked at approximately ten days after onset.[25] Furthermore, we found that a higher level of viral load during the later stage of hospitalization and a longer duration of viral shedding were risk factors for RP events. Sustained viral shedding has been found to be associated with antiviral resistance in patients infected with the influenza H7N9 virus.[26,27] On this basis, we speculate that the higher viral load and longer duration of positive test results in

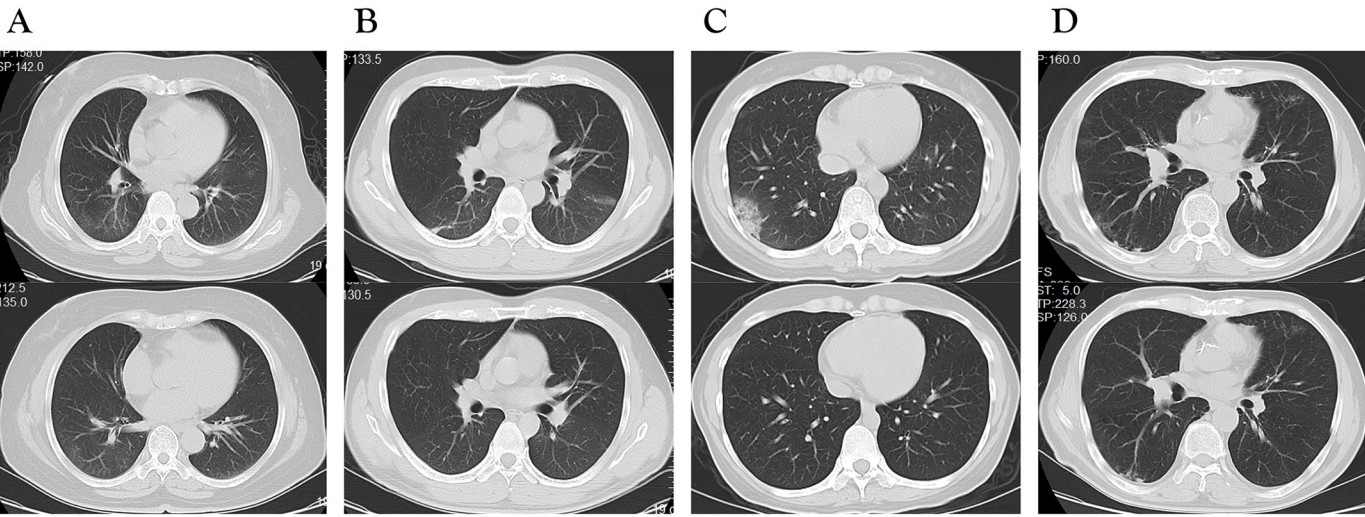

**Fig 4. Chest CT images.** (A) Transverse chest CT images from a 62-year-old woman who got RP 14 days after discharge, showing multiple inflammation in bilateral lungs at readmission (lower panel), which has partly absorbed compared to the condition at first discharged (upper panel). (B) Transverse chest CT images from a 30-year-old man who got RP 8 days after discharge, showing improved multiple inflammation and decreased shadows of fibrotic streaks at readmission (lower panel) compared to the condition at first discharged (upper panel). (C) Transverse chest CT images from a 32-year-old woman who got RP 6 days after discharge, showing inflammation on bilateral lower lobe at readmission (lower panel), which has partly absorbed compared to the condition at first discharged (upper panel). (D) Transverse chest CT images from a 68-year-old male NRP patient, showing multiple inflammation in bilateral lungs 14 days after admission (lower panel), with no obvious change compared with the condition at admission (upper panel).

RP patients may be the consequence of their deficiency in control of viral replication and anti-viral resistance. This makes them more susceptible to positive result after discharged. Since many RP patients shifted from the negative to positive again during their second hospitalization, as they performed at their first discharge. This phenomenon may be caused by the lack of efficiency in virus clearance in RP patients. It was observed these patients had returned to true negative soon but their potential infectiousness remained unknown; thus, prolonged hospitalization for patients with lower efficiency in virus clearance might be a safer measure.

This study has several limitations. First, it's a single-centre study. However, by including adult patients with diverse characteristics, we believe our study population is representative. Second, the estimated duration of viral shedding is limited by the frequency of nasopharyngeal swab samples collection. Third, due to the limited duration of observation, the evaluation for further clinical progression in RP patients could not be carried out, which need a long-term follow-up. Fourth, despite no individual was infected by RP patients in this study, as post-discharge patients were under isolation, we were unable to effectively assess the infectiousness of RP patients.

In conclusion, a nearly 10% incidence of RP events observed in this study suggests numerous COVID-19 patients in the world may get RP. It is expected these patients would return to true negative soon, and unlikely they would get reinfection and remain infectious. We found that a prolonged duration of viral shedding during first hospitalization was a risk factor for RP events which may provide implication on further virological research. However, the clinical symptoms shown in elder RP patients at readmission should not be ignored, suggesting more post-discharge clinical attention on elder COVID-19 patients.

## Ethics approval and consent to participate

This study was approved by the institutional ethics board of Guangzhou Eighth People's Hospital and the requirement for informed consent was waived by the ethics board.

**Table 4. Univariable and multivariable analysis of risk factors associated with RP events.**

|  | Univariate OR | p value | Adjusted OR* | p value |
|---|---|---|---|---|
| **Basic information** |  |  |  |  |
| Age, years | 0.99 (0.97–1.02) | 0.458 | .. | .. |
| Male (*vs.* female) | 0.98 (0.44–2.17) | 0.959 | .. | .. |
| Clinical severity |  |  |  |  |
| Mild | 1 (reference) |  | 1 (reference) |  |
| Moderate | 0.65 (0.18–2.37) | 0.522 | 0.72 (0.19–2.69) | 0.632 |
| Comorbidity |  |  |  |  |
| Any comorbidity | 0.91 (0.38–2.17) | 0.832 | 0.89 (0.12–5.02) | 0.711 |
| Diabetes | 0.39 (0.05–3.03) | 0.370 | .. | .. |
| Hypertension | 1.35 (0.52–3.54) | 0.539 | .. | .. |
| Liver diseases | 0.90 (0.20–4.08) | 0.894 | .. | .. |
| **Laboratory findings** |  |  |  |  |
| Median Ct value (N gene) | 0.96 (0.87–1.07) | 0.496 | 0.96 (0.86–1.07) | 0.487 |
| Week1 | 0.96 (0.87–1.05) | 0.367 | 0.96 (0.87–1.06) | 0.385 |
| Week2 | 0.95 (0.84–1.07) | 0.380 | 0.94 (0.82–1.08) | 0.368 |
| Week3 | 0.85 (0.71–1.03) | 0.103 | 0.88 (0.70–1.10) | 0.256 |
| Median Ct value (ORF gene) | 0.91 (0.82–1.01) | 0.071 | 0.89 (0.80–0.99) | 0.042 |
| Week1 | 0.93 (0.84–1.04) | 0.193 | 0.93 (0.83–1.03) | 0.167 |
| Week2 | 0.90 (0.79–1.04) | 0.144 | 0.87 (0.75–1.02) | 0.078 |
| Week3 | 0.76 (0.60–0.97) | 0.030 | 0.69 (0.50–0.95) | 0.022 |
| Eosinophil,$\times 10^9$/L | 0.84 (0.63–1.11) | 0.213 | 1.59 (0.24–10.75) | 0.633 |
| Week1 | 9.42 (0.07–54.09) | 0.374 | 9.30 (0.06–49.06) | 0.390 |
| Lactate dehydrogenase, U/L | 0.99 (0.99–1.00) | 0.133 | 0.99 (0.99–1.00) | 0.165 |
| Week1 | 0.99 (0.99–1.00) | 0.065 | 0.99 (0.99–1.00) | 0.072 |
| Week3 | 0.99 (0.98–1.00) | 0.122 | 0.99 (0.98–1.00) | 0.193 |
| C-reactive protein, mg/L | 0.98 (0.93–1.02) | 0.303 | 0.98 (0.93–1.03) | 0.392 |
| Week1 | 0.97 (0.94–1.01) | 0.102 | 0.97 (0.94–1.01) | 0.108 |
| **Clinical course** |  |  |  |  |
| Duration of viral shedding from admission, days |  |  |  |  |
| N gene |  |  |  |  |
| ≤10 | 1 (reference) |  | 1 (reference) |  |
| >10 | 5.49 (2.39–12.62) | <0.001 | 5.82 (2.50–13.57) | <0.001 |
| ORF gene |  |  |  |  |
| ≤10 | 1 (reference) |  | 1 (reference) |  |
| >10 | 8.77 (3.64–21.09) | <0.001 | 9.64 (3.91–23.73) | <0.001 |

OR = odds ratio.

*Adjusted for age, sex, hypertension, diabetes and liver disease. OR value in continuous variables is the risk related to per 1 unit increase.

## Supporting information

**S1 Table. Laboratory findings for RP and non-RP patients.** Data of the indicators that have been measured many times are median (IQR) value of first three weeks after admission. Others are data at admission. In the first, second and third weeks, the number of available test results of RP patients were 27, 27, 19, in contrast, 258, 249, 184 in NRP patients. P values comparing RP and NRP patients are from Mann-Whitney U test. GFR = glomerular filtration rate. ALT = Alanine aminotransferase. AST = Aspartate aminotransferase. RP = redetectable as

positive. NRP = non-redetectable as positive.
(DOCX)

**S2 Table. Clinical characteristics, treatment and laboratory findings of 27 RP patients at first admission and readmission.** Data are median (IQR) or n (%). *Results of CT scan for first admission were shown in Table 1. **Only 20 RP patients have tested for Antibody. Results of Ct values and CD cell were median value during hospitalization. RP = redetectable as positive. NRP = non-redetectable as positive.
(DOCX)

**S1 Fig. Entire distribution of times of RP patients.**
(TIF)

## Acknowledgments

We acknowledge the patients, staffs at Guangzhou Eighth People's Hospital, staffs at the Guangzhou Centers for Disease Control and Prevention (CDC) Department of Infectious Disease Control and Prevention, and investigators of the Southern Medical University Department of Epidemiology and Department of Biostatistics for their input and collaboration on this investigation. And we thank the Chinese National Health Commission for coordinating data collection for patients with 2019-nCoV infection; we thank WHO and the International Severe Acute Respiratory and Emerging Infections Consortium (ISARIC) for sharing data collection templates publicly on the website.

## Author Contributions

**Conceptualization:** Jiyuan Zhou, Chunliang Lei, Xianbo Wu.

**Data curation:** Guofang Tang, Furong Li, Huamin Liu, Ziying Yang.

**Formal analysis:** Jiazhen Zheng, Rui Zhou, Furong Li, Yuxin Yuan.

**Funding acquisition:** Xianbo Wu.

**Investigation:** Jiazhen Zheng, Rui Zhou, Fengjuan Chen, Guofang Tang, Keyi Wu, Jianyun Lu.

**Methodology:** Jiyuan Zhou.

**Project administration:** Jiyuan Zhou, Chunliang Lei, Xianbo Wu.

**Resources:** Fengjuan Chen, Guofang Tang, Jianyun Lu, Xianbo Wu.

**Software:** Guofang Tang, Keyi Wu, Furong Li, Yuxin Yuan.

**Supervision:** Huamin Liu, Jiyuan Zhou, Yuxin Yuan, Chunliang Lei, Xianbo Wu.

**Validation:** Huamin Liu, Jianyun Lu, Ziying Yang, Xianbo Wu.

**Visualization:** Keyi Wu, Ziying Yang.

**Writing – original draft:** Jiazhen Zheng, Rui Zhou, Fengjuan Chen.

**Writing – review & editing:** Jiazhen Zheng.

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
