## [Decision Letter · Decision Letter 0]

10 Jul 2020

Dear Professor Wu,

Thank you very much for submitting your manuscript "Incidence, clinical course and risk factor for recurrent PCR positivity in discharged COVID-19 patients in Guangzhou, China: a prospective cohort study" for consideration at PLOS Neglected Tropical Diseases. As with all papers reviewed by the journal, your manuscript was reviewed by members of the editorial board and by several independent reviewers. The reviewers appreciated the attention to an important topic. Based on the reviews, we are likely to accept this manuscript for publication, providing that you modify the manuscript according to the review recommendations. 

Sincerely,

Andrés Felipe Henao-Martínez, M.D.

Deputy Editor

Andrés Henao-Martínez

Deputy Editor

Reviewer's Responses to Questions

**Key Review Criteria Required for Acceptance?**

**Methods**

-Are the objectives of the study clearly articulated with a clear testable hypothesis stated?

-Is the study design appropriate to address the stated objectives?

-Is the population clearly described and appropriate for the hypothesis being tested?

-Is the sample size sufficient to ensure adequate power to address the hypothesis being tested?

-Were correct statistical analysis used to support conclusions?

-Are there concerns about ethical or regulatory requirements being met?

Reviewer #1: Objectives are well documented as the population subgroup study was looking at is positive RP after discharge.

Study design is aptly selected as patient population is prospectively followed and a specific cohort was selected. 

Yes, population was clearly described as only who were positive RP after discharge.

Sample size is small, 27, this is not an adequate power for the hypothesis tested. Having said that, the incidence of RP positive have been around 7 to 10% in previous studies too

Yes correct statistical analysis was used for both continuous and categorical variables. 

Major concern is this being a prospective study, informed consent is required as patient population was tested post discharge and the main criteria for inclusion in the study was RP positive. Manuscript mentions the requirement was waived by ethics board. PLEASE EXPLAIN WHY

Reviewer #2: (No Response)

Reviewer #3: The objectives of the study are clear: analyse epidemiological, virological and clinical data from patients admitted in the hospital and by comparing them with the definitive outcome of retesting positive or not after discharge, identify potential risk factors for restesting positive. It is a perspective study, which is good for its objectives, because it gives to this analysis more power to discover relevant risk factors. 

The characteristic of the populations might be described a little more in depth. Still the authors discarded very few cases and for very clear reasons, so I would deem the population apropriate.

The sample size is large enough to study with enough confidence the problem under consideration.

The statistical analysis are appropriate to test the hypotheses considered.

No ethical concerns (it has all been approved by the responsible ethical committee).

**Results**

-Does the analysis presented match the analysis plan?

-Are the results clearly and completely presented?

-Are the figures (Tables, Images) of sufficient quality for clarity?

Reviewer #1: Yes analysis was appropriate, and results were clearly mentioned. 

Tables did have all the necessary variables.

I can understand the CT images included for three RP patients and one NRP patient. But the relevance and clinical significance need to be explained. Review of literature shows very little correlation between imaging and patient clinical improvement. Even though the findings of eosinophil count LDH appear to have more clinical relevance in follow up too.

Reviewer #2: (No Response)

Reviewer #3: The analysis match the plan and the results are presented very clearly.

The tables and Images are very clear and interesting. In particular figure 3 is very good, gives a very clear visual impact on the point the authors are making.

In figure 2 maybe showing the entire distribution of the times instead of the median might be more informative. This is just a suggestion, it might also be more confusing, still it could be provided as a supplementary figure.

**Conclusions**

-Are the conclusions supported by the data presented?

-Are the limitations of analysis clearly described?

-Do the authors discuss how these data can be helpful to advance our understanding of the topic under study?

-Is public health relevance addressed?

Reviewer #1: Conclusions are partially supported by the data. I agree with conclusions about patient population >60 who are RP positive, viral load predicting disease progression and risk factors identified for being RP positive. 

My concern is with the conclusion that these RP patients unlikely are infectious. Even though a considerable conclusion can be made that they are unlikely reinfected based on provided biochemical data and symptoms, we have no evidence that they are not infectious to other population or health care personnel. Please explain

Limitations have been documented well. Please explain the diverse characteristics of the adult population, which was described in the manuscript that makes the study representative. 

Data or study findings definitely help in better understanding of the pandemic but conclusions are not completely accurate. 

We can only address public health relevance, if the diverse characteristics are explained. Also such a small population with not adequate power is a concern. But in setting of a pandemic, can be considered for further revisions

Reviewer #2: (No Response)

Reviewer #3: The conclusions are supported by the data presented.

I find that overall the discussion is broad and exaustive and very relevant to understand the impact and relevance of this study on the topic.

the public health relevance of the study is clearly addressed.

I have some issues on the limitations they describe: 

- the characteristic of the populations are identified as the cause of discrepancies with previous studies on the same subject, but the authors do not detail which characteristic my affect the difference? Is the average age different? Is the ethnicity different? (line 203-204)

-I do not find 285 and 209 (lines 202-203) to be so different as sample sizes and the authors did not detail which population characteristics differ among the three studies. Can the authors comment on this with more detail? 

- How does this number of redetectable positives compare with the specificity of the rt-pcr test they used?

- Since some of the authors are directly responsible for performing the nasal-throat swab on patients, could they comment on how the way the nasal-throat swab is performed might affect (if at all) their results? Would a sputum-based test impact in any way on their results?

**Editorial and Data Presentation Modifications?**

Reviewer #1: 1) Explain conclusion that these RP patients unlikely are infectious.

2) diverse characteristics of the adult population

3)explain informed consent process

4)

Reviewer #2: (No Response)

Reviewer #3: I have a couple of general issues that are unclear for researchers not completely familiar with the local response to CoVid-19. I would suggest the authors comment more in their manuscript:

1. Why is there patients follow up after discharge? Is it a feature added from this study or is it common practice? If it is a feature of this study they should detail how the frequency of re-testing was decided.

2. Why if the discharged patients re-test positive they are re-admitted into the hospital? Is it common practice in their region? Is it a feature added by this specific study? 

3. How does this number of redetectable positives compare with the specificity of the rt-pcr test they used?

4. Is there any possibility that any of the included patients has had a RP after the end of the study? They should clearly explain why this is not possible (i.e. all the patients have been discharged and checked for more than x weeks and it is unlikely to have a RP after this time).

In figure 2 maybe showing the entire distribution of the times instead of the median might be more informative. This is just a suggestion, it might also be more confusing, still it could be provided as a supplementary figure.

**Summary and General Comments**

Reviewer #1: Study strengths

1) design of study and data provided

2) able to define incidence, risk factors for RP positive, and clinical characteristics

3) population selection and manuscript writing

Weakness have been defined above. Sample size, power and generalizability to society are concerns

One other major concern is the availability of Ct values described and the test availability in most nations. 

Study has significance in the present pandemic situation. Study has novelty in the number of tests done and follow up period. 

Ethical concerns as above with informed consent

Reviewer #2: Results:

1. Table 2 Normal range for CT value ≥40 needs to be defined, as per my understanding it should be ≤40, if the authors mean something else they should explain it in the footnotes.

2. There is a difference in the viral shedding days between the N gene and ORF gene, can the authors postulate a probable cause for this phenomenon.

3. Line 168 results section mentions about the immunological assay (IgG and IgM) theses need to be defined in the methodology (test kit used, type of assay I,e, ELISA based or ICT).

4. Table 2 shows a p value of 0.045 for difference in Ct value at 3rd week however the results in table 4 are completely different. If both are comparison between RP and NRP then why this discrepancy. If these are two different analysis the title of the table should define it accordingly. 

Discussion:

1. Line 198-202 the language needs to be modified it sounds a bit confusing

2. Line 203- “The incidence rate of our study is representative due to the diverse characteristics of the study population”. What characteristics are the authors talking about?

3. The author’s need to provide a justification or their view on why the patients who were negative at time of discharge after first hospitalization became positive? Is there any limitation of the test (in view of the Limit of detection) or is there a possibility that these might be case of reinfection?. 

4. The justification produced to rule out reinfection completely is not reasonable, understanding that many cases now come as asymptomatic and knowing that the antibodies might have some effect on the virus the re-infected patients might present with mild symptoms, the authors can presume that these were not re-infection cases but the study methodology is not sufficient to rule out the same.

Reviewer #3: I think this manuscript is good enough to deserve pubblication with minor changes that I detailed above.

One of its major strength is that it is a perspective study and very few patients have been discarded and for very limited reasons (death and <18 years of age), so the study is significative, in that adds new knowledge to the field. The study evaluates a range of different epidemiological, clinical and virological variables and how they influence the re-positive testing outcome using appropriate statistical tests.

Some parts should be clearer, like what is part of the study design and what is rutine in Guangzhou Eighth People's Hospital (Guangzhou, Guangdong). For example is the follow-up testing after discharge rutin or part of the study? Is readmission to hospital upon positive re-testing rutine or part of the study design?

Every study of this kind is novel, in the sense that explores a new population and a slighltly different set of variables. Also this study has a slightly bigger sample size compared to the previous ones.

It is significant in that it gives risk factors to predict patients that might re test positive after testing negative. This is importat for example to exclude re-infections. But also to help any national healthcare system to be prepared to such cases (and how many to expect).

As a general execution it is well conceived, executed and written. They also provide all the data in tables that are very easy to read.

PLOS authors have the option to publish the peer review history of their article (what does this mean?). If published, this will include your full peer review and any attached files.

Reviewer #1: Yes: Pradeep Yarra

Reviewer #2: No

Reviewer #3: Yes: Alice Ledda
---

## [Editor Report · Decision Letter 1]

27 Jul 2020

Dear Professor Wu,

We are pleased to inform you that your manuscript 'Incidence, clinical course and risk factor for recurrent PCR positivity in discharged COVID-19 patients in Guangzhou, China: a prospective cohort study' has been provisionally accepted for publication in PLOS Neglected Tropical Diseases.

Best regards,

Andrés Felipe Henao-Martínez, M.D.

Deputy Editor

---

## [Editor Report · Acceptance letter]

25 Aug 2020

Dear Professor Wu,

We are delighted to inform you that your manuscript, "Incidence, clinical course and risk factor for recurrent PCR positivity in discharged COVID-19 patients in Guangzhou, China: a prospective cohort study," has been formally accepted for publication in PLOS Neglected Tropical Diseases.

Best regards,

Shaden Kamhawi

co-Editor-in-Chief

Paul Brindley

co-Editor-in-Chief
